# Characterization of the Binding Modes of Cu^2+^ Ions with Tyrosine and Ado, AMP, ADP, and ATP: A Comprehensive Potentiometric, Spectroscopic, and Computational Approach

**DOI:** 10.3390/ijms26188865

**Published:** 2025-09-11

**Authors:** Patrycja Sadowska, Romualda Bregier-Jarzębowska, Wojciech Jankowski, Mateusz R. Gołdyn, Renata Jastrząb

**Affiliations:** 1Faculty of Chemistry, Adam Mickiewicz University in Poznań, Uniwersytetu Poznańskiego 8, 61-614 Poznań, Poland; patrycja.sadowska@amu.edu.pl (P.S.); bregier@amu.edu.pl (R.B.-J.); wojciech.jankowski@amu.edu.pl (W.J.); mateusz.goldyn@amu.edu.pl (M.R.G.); 2Center for Advanced Technologies, Uniwersytetu Poznańskiego 10, 61-614 Poznań, Poland

**Keywords:** coordination compounds, copper(II) complexes, tyrosine, purine nucleotide ligands, potentiometric analysis, spectroscopic characterization, molecular simulations and DFT calculations

## Abstract

We report the mode of interaction of copper(II) ions with tyrosine (Tyr, L) in binary and ternary systems with Ado, AMP, ADP, and ATP (L’) as second ligands in an aqueous solution. The composition and overall stability constants of the complexes formed were determined using the potentiometric method. The coordination sites were identified through spectroscopic (VIS, EPR, IR) methods, as well as DFT and computational–molecular modeling. In the binary Cu(II)/Tyr system, the main reaction centers of the ligand molecule involved in the interactions with copper(II) ions are nitrogen (-NH_2_ group), as well as oxygen atoms (-COO^−^ group), as confirmed, for example, by comparing the mode of coordination in the CuH_2_(Tyr)_2_ species and the [CuH_2_(Tyr)_2_(H_2_O)] × 1.5H_2_O solid complex obtained. In the ternary Cu(II)/L/L’ systems, MLHxL’ and mixed MLL’ protonated complexes are formed. Only in the ATP system were no MLL’(OH)_x_ hydroxocomplexes found. An increase in the number of phosphate groups in ADP and ATP molecules has no effect on their participation in the coordination in ternary species, and these ligands interact just like in binary species (i.e., in ADP, both *α*- and *β*-phosphate groups, and in ATP, only the *γ*-phosphate group). It was observed that the introduction of a second ligand into the Cu(II)/Tyr system did not change, over the entire pH range studied, the tyrosine coordination mode.

## 1. Introduction

The study of transition metal ion complexes with ligands of biological interest is of ongoing interest due to their important role in many physiological and biochemical processes. Particularly valuable are model systems, the characterization of which allows for a better understanding of the mechanisms of the binding and selectivity of metal–ligand interactions [1,2,3].

One of the interesting bioligands in this type of system is tyrosine (Tyr; Figure 1). Tyrosine belongs to the 20 natural amino acids and is a precursor of the catecholamine neurotransmitters dopamine, epinephrine, and norepinephrine [4,5], as well as the hormones thyroxine and triiodothyronine [6]. This bioligand is a constituent of many protein blocks [7]; participates in metalloenzyme-catalyzed substrate oxidation [8]; and forms complexes with biological activity against some bacteria and fungi [9]. Additionally, the presence of tyrosine is crucial for electrical transfer in many biomolecules, e.g., phototropin, cryptochromes, and ribonucleotide reductase [10], and its shortage in the human body could be a cause of depression or chronic fatigue [5].

Also worth mentioning are nucleosides and nucleotides—fragments of nucleic acids of vital importance to living organisms [11,12,13]. The main tasks of these multifunctional biomolecules are not only storage and transfer of genetic information but also intramolecular energy transfer and synthesis of lipids [11,14]. Adenosine (Ado; Figure 1) as well as adenine nucleotides (AMP, ADP, ATP; Figure 1) are found in all animal organ systems [15]. Adenosine is an endogenous nucleoside that plays a crucial role in cellular energy production across all living organisms and regulates many physiological functions, including immune responses and neuronal activity [16,17,18]. In turn, adenosine 5’-monophosphate (AMP), adenosine 5’-diphosphate (ADP), and adenosine 5’-triphosphate (ATP) are considered the main energy stores in cells [19]. Moreover, among the related nucleotides, AMP is known for its role in maintaining energy homeostasis and activating protein kinase enzymes [19,20]; ADP is key in mediating vascular and cellular signaling [21,22]; and ATP functions as a neurotransmitter in both the peripheral and central nervous systems [23,24].

It is currently known that metal ions play crucial roles in living organisms; for example, they are involved in the transport of oxygen, enzymatic reactions, and stabilizing the DNA structure [25,26]. Metal complexes which form between positively charged metal centers and negatively charged biomolecules (such as amino acids or nucleotides) could be interesting as model compounds for a better understanding of the interactions that take place in biological systems [8,27]. Among them, copper(II) ions play a significant role as cofactors for some enzymes [4] and are required for the proper functioning of the brain [26] and growth and development in many organisms [23]. Additionally, it is extremely important to maintain proper levels of Cu(II) ions in the human body because an abnormal copper metabolism or content leads to various diseases [28]. Moreover, copper(II) complexes are known to have diagnostic and therapeutic potential in treating human diseases [29] and can be applied as, e.g., antimicrobial, antitumor, or antiviral agents [30]. Many studies have shown that copper(II) complexes can interact with DNA [9]—for example, non-covalently—providing a flexible framework for many biological processes to occur [12,31].

Although the ability of tyrosine and adenine nucleotides to form binary as well as ternary complexes with metal ions is known in the literature [32,33,34,35,36], there is a lack of studies describing the interactions in ternary systems considered in terms of this work. Moreover, we are focusing on this topic to gather new experimental data on their interactions under conditions that mimic biological systems. This approach also allows us to update existing information in the literature and consequently improve our understanding of the processes occurring within living organisms. In this study, we aimed to determine the mode of coordination complexes formed in the Cu(II)/tyrosine system and answer questions about the interactions in ternary Cu(II)/tyrosine/adenosine or adenosine phosphate systems.

## 2. Results and Discussion

In a tyrosine molecule, the potential coordination centers are the carboxyl group, the phenolic hydroxyl group, and the amino group. The order of proton dissociation follows the typical pattern for amino acids with ionized groups. At the beginning, under acidic conditions, a proton dissociates from the carboxyl group. As the pH value increases, the amino group dissociates, followed by the hydroxyl phenolic [37,38]. The tyrosine protonation constants and overall stability constants (log*β*) of the compounds formed in the binary system Cu(II)/Tyr have been reported in the literature previously [38,39,40,41,42,43]. The results obtained in our studies are in good agreement with those in the literature and were used to study the mixed ternary complexes. In Ado, AMP, ADP, and ATP molecules, the potential metal ion binding centers are endocyclic N(1) or N(7) nitrogen atoms from the purine ring, as well as phosphate groups from the nucleotides [44,45] (Figure 1). The sequence of adenosine nucleotide protonation is as follows: log*K_e_* of about 6.5—the protonation of the phosphate group; log*K_e_* of about 4.0—the protonation of the nitrogen atom; and log*K_e_* of 0.4, 1.02, and 1.7—the second site at the phosphate group in H_3_(AMP), H_3_(ADP), and H_3_(ATP), respectively [11,14]. Despite the higher basicity of the N(1) atom in the adenosine ring compared to that of N(7), the latter is the favored site for metal coordination [46,47]. Both the values of the protonation constants of these ligands and the stability constants (log*β*) of the complexes in Cu(II)/Ado, Cu(II)/AMP, Cu(II)/ADP, and Cu(II)/ATP systems were determined earlier by our group [44,48,49,50] and, as for tyrosine, were used in the analysis of the ternary systems.

### 2.1. Interactions in the Binary Cu(II)/Tyrosine System

In the Cu(II)/tyrosine system studied, potentiometric measurements were carried out at metal-to-ligand ratios of 1:1, 1:2, and 1:4. For each of these molar ratios, potentiometric titration data were analyzed using the HYPERQUAD 2008 program. Due to the formation of the largest number of species with the lowest standard deviations in their stability constants (log*β*), as well as the value Ʃ, only the system with the ratio M:L = 1:2 was selected for further consideration. Moreover, for the chosen complexes formed in the investigated system, quantum-chemical calculations were performed, and on the basis of the results obtained, the most energetically favorable modes of interaction were determined. In the analyzed system, we found the formation of the four complexes, i.e., CuH_2_(Tyr), CuH(Tyr), CuH_2_(Tyr)_2_, and CuH(Tyr)_2_. The values determined of the overall stability constants (log*β*) as well as the equilibrium constants (log*K_e_*) of these complexes are presented in Table 1. The values obtained are in the good agreement with the literature [38,39,40,41,42,43].

The selection of appropriate coordination models was supported by the strong agreement between the experimental titration curves and those obtained via computer simulation, which took into account all relevant species. Furthermore, the noticeable deviation in the experimental titration curve from that for free tyrosine above a pH of 3.0 indicates the onset of complex formation at this pH value (Figure 2a).

With a pH from ~2.5 to ~5.5, the formation of a CuH_2_(Tyr) complex (log*β* = 22.39) is observed, which dominates at a pH close to 3.0 (Figure 2b). The values of the spectroscopic parameters obtained from the VIS and EPR spectra at a pH of 3.0 are λ_max_~780 nm (Figure 3a) and g_||_ = 2.349, A_||_ = 140 × 10^−4^ cm^−1^ (Table 2), suggesting the involvement of a deprotonated carboxyl group (-COO^−^) from tyrosine in the coordination with copper(II) ions (chromophore {O} type) [52].

With an increasing pH, another protonated complex is formed, i.e., CuH(Tyr) (log*β* = 18.54), which dominates from a pH of about 4.0 to 5.7, binding more than 70% of the Cu(II) ions at its maximum concentration (Figure 2b). The VIS and EPR spectroscopic analysis (Figure 3b, Table 2) suggests that the Cu(II) ion is coordinated through the oxygen atom of the carboxylate group and the nitrogen atom of the amino group, the chromophore {1N,1O} type. Moreover, comparison of the values of log*K_e_* = 8.26 for CuH(Tyr) with log*K_e_* = 3.10 for CuH_2_(Tyr) ({O}) confirms the participation of another active center in the interactions in the monoprotonated form. Unfortunately, no ML-type species were found in this system, as was also the case in other papers [41,42,43]. In the pH range of 5.7–9.3 (Figure 2b), a CuH_2_(Tyr)_2_ complex occurs, which is formed according to the equation CuH(Tyr) + H(Tyr) ⇌ CuH_2_(Tyr)_2_. An analysis of the VIS spectra for this species indicates a shift in the d-d band towards higher energies (pH = 7.8, λ_max_ = 620 nm), relative to its position for CuH(Tyr) (pH = 4.8, λ_max_ = 670 nm), which corresponds to the involvement of another centers, i.e., nitrogen and oxygen atoms, in metalation, the chromophore {2N,2O} type [53]. Such a blue shift in the d–d band, accompanied by a decrease in g_∥_ (from 2.307 to 2.257, Table 2), is a concerted change, typically associated with strengthening of the equatorial ligand field in Cu(II) complexes and is therefore consistent with the rearrangement of the coordination environment. This is confirmed by the EPR parameters, too (Table 2). The equilibrium constant log*K_e_* = 6.56 of the attachment of the second H(Tyr) molecule to the anchoring CuH(Tyr) is lower than that for CuH(Tyr) (log*K_e_* = 8.26), which is due to the spatial conditions and may indicate its different coordination symmetry for this ligand molecule. Moreover, a solid CuH_2_(Tyr)_2_ complex was obtained, for which the formula [CuH_2_(Tyr)_2_(H_2_O)] × 1.5H_2_O was determined. This compound occurred in the form of blue crystals suitable for an X-ray analysis Appendix A. Although the crystal structure of an analogous complex was described earlier [53], our compound was obtained using a different crystallization method, and its crystallographic parameters were slightly better Appendix A. The FT–IR spectrum of the [CuH_2_(Tyr)_2_(H_2_O)] × 1.5H_2_O complex and a comparison with free tyrosine in the solid state are shown in Appendix A. In the spectrum of the complex, the bands observed in the range 1700–1300 cm^−1^, the most interesting from a coordination point of view, are shifted in relation to those for free tyrosine, indicating complex formation, and are in the good agreement with the literature [53]. Moreover, as was described earlier, the frequency region below 600 cm^−1^ can provide information on Cu(II)−ligand vibrations [54,55]. The FT−IR band at 419 cm^−1^ is assigned to the ν(Cu−N_Tyr_) stretching vibrations and indicates the participation of the amino group in the metal coordination. At a pH of 10.2, the predominant complex is CuH(Tyr)_2_, binding almost 80% of the copper(II) ions. The pH range for the occurrence of this form indicates that it can be formed through the attachment of free tyrosine to the CuH(Tyr) binary complex. The value of log*K_e_* = 7.45 for this reaction, comparable with the log*K_e_* values corresponding to CuH(Tyr) and CuH_2_(Tyr) species (Table 1), suggests that deprotonated tyrosine, similarly to H(Tyr), interacts with copper(II) ions with the participation of the chromophore {1N,1O} type. Moreover, VIS and EPR studies (Table 2, Figure 3a) indicate that the mode of coordination in this species is similar that for CuH_2_(Tyr)_2_.

#### Quantum-Chemical Calculations in the Binary Cu(II)/Tyr System

Preliminary geometry optimizations showed that for complexes involving one molecule of monoprotonated tyrosine with a copper(II) ion, only CuH(Tyr)_1 is likely to be observed. CuH(Tyr)_2 and CuH(Tyr)_3 structures after optimization are converted into CuH(Tyr)_1; see Appendix A. For complexes involving two molecules of H(Tyr) with a copper(II) ion, only one structure is likely to be observed, and it is the CuH_2_(Tyr)_2__1 structure (CuH_2_(Tyr)_2__2 and CuH_2_(Tyr)_2__2 structures are converted into CuH_2_(Tyr)_2__1 after optimizations); see Appendix A. For complexes consisting of H(Tyr) and Tyr molecules with a copper(II) ion, only two structures are likely to be observed: CuH_2_(Tyr)_2__1 and CuH_2_(Tyr)_2__2, (CuH_2_(Tyr)_2__3 structure after optimization is converted into CuH_2_(Tyr)_2__1); see Appendix A. Energy calculations for complexes consisting of one molecule of H(Tyr) and a copper(II) ion were conducted. The calculated energies, monomer sums, and interaction energies in an aqueous solution are presented in Appendix A. These results indicate that the interaction energy was −176.6 kcal/mol. In this structure, tyrosine likely coordinates the copper ion via oxygen and nitrogen atoms. The optimized structure is presented in Figure 4, with the atomic coordinates provided in the Appendix A.

Calculations for complexes consisting of two H(Tyr) molecules and a copper(II) ion were also performed. The calculated energies, sum of monomers, and interaction energies in water are gathered in Appendix A. As shown in Appendix A, the Cu_H_2__(Tyr)_2__2 interaction energy was −271.3 kcal/mol. In this complex, both molecules of tyrosine coordinate cooperatively with the oxygen and nitrogen atoms. The optimized structure of this complex is depicted in Figure 5, and coordinates are included in Appendix A.

Similarly, calculations were performed for complexes consisting of H(Tyr) and Tyr molecules with a copper(II) ion. The calculated energies, sum of monomers, and interaction energies in water are presented in Appendix A. The results show that the strongest interaction energy among these complexes occurs in the Cu_H_(Tyr)_2__1 scheme, with an interaction energy of −274.1 kcal/mol. In this complex, H(Tyr) and Tyr molecules coordinate with the copper(II) ion via oxygen and nitrogen atoms from both molecules. The optimized structure of this complex is depicted in Figure 6, and the coordinates of both obtained structures are included in Appendix A.

### 2.2. Interactions in Cu(II)/Tyrosine/Adenosine (or AMP, ADP, ATP) Ternary Systems

In the ternary systems studied, potentiometric titration was carried out at metal–ligand1–ligand2 molar ratios of 1:1:1 and 1:2:2. We selected equimolar systems for the subsequent analysis, considering the same criteria as those in the binary system, specifically the formation of the highest number of complex species and the lowest standard deviation in the overall stability constants (log*β*). To determine the overall stability constants of the complexes formed in the ternary Cu(II)/Tyr/Ado or AMP, ADP, and ATP systems, we took into account the values of the protonation constants of the ligands, as well as the overall stability constants (log*β*) of the complexes formed in the binary Cu(II)/Tyr, Cu(II)/Ado, Cu(II)/AMP, Cu(II)/ADP, and Cu(II)/ATP systems (Table 3) [44,48,49,50]. We found no reports on spectroscopic and potentiometric studies of ternary Cu(II)/Tyr/Ado or AMP or ADP systems, except for a Cu(II)/Tyr/ATP system (using different conditions and a different set of complexes) [56]. In all analyzed systems, protonated MLH_x_L’ as well as MLL’ complexes are formed. Solely in the Cu(II)/Tyr/ATP system was the formation of the MLL’(OH)_x_ forms not detected.

#### 2.2.1. Cu(II)–Tyrosine–Adenosine Complexes

Figure 7 presents the distribution and VIS spectra of species forming in the analyzed system. The first Cu(Tyr)H_3_(Ado) species is present in the system within a pH range from 2.7 to about 5.0, where Ado is fully protonated and the binary complex CuH_2_(Tyr) is also present.

The Cu(Tyr)H_3_(Ado) complex forms according to the reaction CuH_2_(Tyr) + H(Ado) ⇌ Cu(Tyr)H_3_(Ado). As was reported previously [58], there is a relationship between the number and type of donor atoms in the inner coordination sphere of Cu(II) and the energy of the d-d transition. The value of this transition for the Cu(Tyr)H_3_(Ado) complex λ_max_ ~790 nm is comparable to λ_max_ = 780 nm for CuH_2_(Tyr), which suggests a similar mode of interaction of H_2_(Tyr) with Cu^2+^ ions in this ternary species. In turn, adenosine (with the donor center blocked) is located outside of the coordination sphere of the metal ions and takes part in non-covalent interactions with the anchoring binary CuH_2_(Tyr) species, forming a molecular complex. This type of interaction is also in accordance with the EPR parameters, g_||_ = 2.402, A_||_ = 130 × 10^−4^ cm^−1^, g_┴_ = 2.069, and A_┴_ = 5 × 10^−4^ cm^−1^ ({1O} chromophore) [52]. With the deprotonation of the Ado molecule, a diprotonated Cu(Tyr)H_2_(Ado) complex is formed, which has its maximum concentration at a pH ~4.0. The inclusion of another active center in the coordination in this species, i.e., the nitrogen atom from the purine ring of Ado, shifts the maximum absorption towards shorter wavelengths relative to those for Cu(Tyr)H_3_(Ado) (Table 4), indicating interactions with the chromophore {1N,1O} type [59]. Moreover, this mode of coordination is confirmed by EPR spectroscopic studies, i.e., g_||_ = 2.370, A_||_ = 120 × 10^−4^ cm^−1^, g_┴_ = 2.066, and A_┴_ = 8 × 10^−4^ cm^−1^; see Figure 8a.

Another form is formed with the deprotonation of the -NH_3_^+^ group from the tyrosine molecule, CuH(Tyr) + Ado ⇌ Cu(Tyr)H(Ado). The spectroscopic parameters (Table 4) indicate the involvement of the oxygen atom and one nitrogen atom, chromophore {1N,1O}, in the interactions. Interestingly, this proves that the nitrogen atom from the amino –NH_2_ group of the tyrosine molecule is not involved in the coordination. Therefore, in both the Cu(Tyr)H_2_(Ado) and Cu(Tyr)H(Ado) complexes, the same mode of metalation is observed. From a pH ~6.5, the Cu(Tyr)(Ado) species occurs and dominates at a pH close to 8.8. The values of the spectroscopic parameters, at this pH, are λ_max_ = 635 nm, g_||_ = 2.251, A_||_ = 185 × 10^−4^ cm^−1^, g_┴_ = 2.055, and A_┴_ = 5 × 10^−4^ cm^−1^ and correspond to the chromophore {2N,1O} type [60]. This suggests that in addition to the nitrogen atom from Ado, a nitrogen atom (-NH_2_ group) and an oxygen atom (-COO^−^ group) from the tyrosine molecule are also involved in the coordination. Moreover, the oxygen atom from the deprotonated -OH group of tyrosine is not engaged in an interaction. This mode of coordination is confirmed using quantum mechanical calculations (see Section 2.3.1). Above a pH of 10.0, in the system, the Cu(Tyr)(Ado)(OH) hydroxocomplex predominates, which is formed upon the attachment of a water molecule to MLL’ according to Cu(Tyr)(Ado) + H_2_O ⇌ Cu(Tyr)(Ado)(OH) + H^+^. In this form, similar coordination of the ligands with the Cu(II) ions as that in the MLL’ form is observed. Slight differences in the VIS and EPR data indicate the involvement of an oxygen atom from the -OH group of H_2_O in the inner coordination sphere of metal ions (Table 4, Figure 8b).

#### 2.2.2. Cu(II)–Tyrosine–Adenosine-5’-Monophosphate Complexes

The Cu(Tyr)H_3_(AMP) complex forms in a pH range of 2.5–5.5 (Figure 9a). On the basis of the order of proton dissociation of both the Tyr and AMP molecules, it was concluded that in this species, the oxygen atom from the –COO^−^ group of tyrosine is involved in the coordination. Meanwhile, it is difficult to determine which of the deprotonated donor centers of AMP is directly involved in the coordination. Unfortunately, Cu(Tyr)H_3_(AMP), similarly to Cu(Tyr)H_2_(AMP) species, forms in a pH range in which significant concentrations of CuH_2_(Tyr) and CuH(Tyr) species are observed; therefore, Vis and EPR spectroscopic studies could not be carried out. With the complete deprotonation of AMP, Cu(Tyr)H_2_(AMP) is formed according to CuH_2_(Tyr)_2_ + AMP ⇌ Cu(Tyr)H_2_(AMP).

A monoprotonated species forms at a pH of about 4.0 according to the reaction CuH(Tyr) + AMP ⇌ Cu(Tyr)H(AMP). The results of electronic (Figure 9b) and EPR Appendix A spectrum analyses at a pH = 6.6 (Table 4), at which this complex dominates, suggest a {2N,2O}-type chromophore [59]. Comparison of λ_max_ = 670 nm for Cu(Tyr)H(AMP) with λ_max_ = 700 nm for Cu(Tyr)H(Ado), the {1N,1O} chromophore, may indicate the inclusion of another center of the ligand in the interaction with the Cu(II) ions, which is supported by molecular modeling (see Section 2.3.2). With an increasing pH and complete deprotonation of the Tyr molecule, the formation of Cu(Tyr)(AMP) is observed. The spectroscopic VIS and EPR data for the MLL’ complex (λ_max_ = 650 nm, g_||_ = 2.300, A_||_ = 174 × 10^−4^ cm^−1^) are in good accordance with those for the monoprotonated form, which indicates a similar mode of coordination in both of these species, i.e., the chromophore {2N,2O} type [59]. The mode of interaction in Cu(Tyr)(AMP) was also confirmed through quantum-mechanical calculations. Starting with a pH close to 9.0, the formation of Cu(Tyr)(AMP)(OH) begins and it becomes dominant above a pH of 10.5. The changes in the maximum absorption, as well as the EPR parameters, relative to the MLL’ species (Table 4) confirm the involvement of the oxygen atom from–OH in the water molecule in the coordination.

#### 2.2.3. Cu(II)–Tyrosine–Adenosine-5’-Diphosphate Complexes

Computer analysis of the potentiometric data showed that in the system studied, at a pH below 3.0 (Figure 10a), the Cu(Tyr)H_3_(ADP) complex appears. Within the same pH range, in binary systems, CuH_2_(Tyr) and CuH(ADP), as well as H(ADP) [44], form. The equilibrium constant for Cu(Tyr)H_3_(ADP) log*K_e_* = log*β*_Cu(Tyr)H3(ADP)_ − log*β*_CuH2(Tyr)_ − log*β*_H(ADP)_ = 4.71 is in good accordance with log*K_e_* = 4.29 for CuH(ADP), which could suggest a similar mode of coordination of the ADP molecule in binary and ternary species, i.e., {1N,1O} [44]. Unfortunately, for Cu(Tyr)H_3_(ADP), similarly to Cu(Tyr)H_2_(ADP), the concentrations of binary forms in the pH range of ternary species formation were high enough to make a spectroscopic analysis impossible.

In turn, a diprotonated Cu(Tyr)H_2_(ADP) species is observed from a pH of 2.6 to ~6.0 (Figure 10a). The value of log*K_e_* for this complex, calculated according to log*K_e_* =log*β*_Cu(Tyr)H2(ADP)_ − log*β*_CuH2(Tyr)_ = 7.34, is comparable with log*K_e_* = 6.99 for Cu(ADP) [44], which could indicate an analogous type of metalation, with the oxygen atom from the -COO^−^ group of Tyr, the nitrogen atom from the purine ring, and the oxygen atoms from the phosphate groups of the ADP. With an increasing pH and deprotonation of the nitrogen atom (-NH_2_) of Tyr, the Cu(Tyr)H(ADP) species is formed (Figure 10a) through the reaction CuH(Tyr) + ADP ⇌ Cu(Tyr)H(ADP). The value of log*K_e_* = 6.57 for this complex is similar to that for Cu(Tyr)H_2_(ADP) and proves an analogous mode of interaction in both complexes. The energy of the d-d transitions in the monoprotonated complex at a pH = 5.6 is λ_max_ = 725 nm, which indicates the formation of a {1N,xO}-type chromophore. The above conclusion is supported by an analysis of the EPR results: g_||_ = 2.362 and A_||_ = 138 × 10^−4^ cm^−1^
Appendix A [33]. Therefore, it can be concluded that not all of the donor centers of both ligand molecules are directly involved in the coordination. In the pH range of Cu(Tyr)(ADP) formation (Figure 10a), Tyr and ADP are already fully deprotonated. The shifts in the d-d band in the spectra of the resulting solution towards a higher energy compared to that for monoprotonated species (Table 4, Figure 10b), as well as the EPR parameters (g_||_ = 2.325, A_||_ = 146 × 10^−4^ cm^−1^), clearly show that in MLL’, the {2N,3O}-type chromophore is formed [59]. This type of coordination is confirmed using quantum-chemical calculations (see Section 2.3.3).

The Cu(Tyr)(ADP)(OH): Cu(Tyr)(ADP) + H_2_O ⇌ Cu(Tyr)(ADP)(OH) + H^+^ and Cu(Tyr)(ADP)(OH)_2_: Cu(Tyr)(ADP)(OH) + H_2_O ⇌ Cu(Tyr)(ADP)(OH)_2_ + H^+^ species are dominant at pHs of about 9.6 and 10.5, respectively. Shifts in the maximum absorption bands for these hydroxocomplexes by 25 nm and 35 nm, relative to λ_max_ = 675 nm for Cu(Tyr)(ADP), as well as the EPR studies (Table 4) [59,60] suggest the involvement of one or two water molecules into the coordination sphere of the copper(II) ion, respectively. However, the participation of a second water molecule in the coordination removes one of the donor centers from the inner coordination sphere of the copper(II) ion, i.e., the oxygen atom of the Tyr or ADP molecule.

#### 2.2.4. Cu(II)–Tyrosine–Adenosine-5’-Triphosphate Complexes

The Cu(Tyr)H_4_(ATP) species is observed at a strongly acidic pH (Figure 11a). As evidenced by the VIS (λ_max_ ~800 nm, Figure 11b) and EPR (g_||_ = 2.397, A_||_ = 129 × 10^−4^ cm^−1^, Appendix A) spectra for this form, the oxygen atoms from the deprotonated carboxyl group of tyrosine, as well as from the phosphate group of ATP, are engaged in the coordination with Cu(II) ions.

Another complex, i.e., Cu(Tyr)H_3_(ATP) is observed at a pH where a mixture of tetraprotonated as well as diprotonated forms also occurs at high concentrations (Figure 11a). For this reason, VIS and EPR spectroscopic studies were not performed, and consequently, it was not possible to clearly determine the mode of interaction in this species. However, Cu(Tyr)H_3_(ATP) is formed in accordance with the reaction CuH_2_(Tyr) + H(ATP) ⇌ Cu(Tyr)H_3_(ATP), which suggests that only the oxygen atom from the -COO^−^ group of tyrosine and the oxygen and nitrogen atoms from the purine ring of ATP can participate in the metalation. With an increasing pH and the complete deprotonation of ATP, the Cu(Tyr)H_2_(ATP) complex is formed, which binds 60% of the copper(II) ions at a higher pH (Figure 11a). An analysis of the spectroscopic data at a pH = 5.2 (λ_max_ = 760 nm, g_||_ = 2.363, and A_||_ = 147 × 10^−4^ cm^−1^) indicates the formation of the {1N,1O}-type chromophore, and they are in the good agreement with the results obtained for analogous coordination systems [61,62]. At this pH, the -NH_3_^+^ and -OH groups of tyrosine are blocked from the interaction, and hence, only the oxygen atom from the -COO^−^ group of this ligand can participate in the coordination. This indicates that ATP coordinates with the copper(II) ion in this ternary species, differently from the Cu(ATP) complex ({1N,1O} [50]), i.e., with the participation of only the nitrogen atom from the purine ring, while the phosphate groups are outside the coordination sphere. A monoprotonated Cu(Tyr)H(ATP) species is formed at a pH above 4.0 (Figure 11a). Under the conditions of this complex dominating, at a pH close to 8.0, the NH_3_^+^ group of Tyr becomes deprotonated and can participate in metal coordination. The shift in the maximum absorption band towards shorter wavelengths λ_max_ = 705 nm, as well as changes in the EPR parameters (Table 4), relative to those of Cu(Tyr)H_2_(ATP), clearly indicates the involvement of another atom in the interaction with the metal ion with the {2N,2O} chromophore. Which donor centers directly take part in coordination was established on the basis of quantum-chemical studies. With the deprotonation of the –OH group of tyrosine, a Cu(Tyr)(ATP) complex is formed (Figure 11b). Unfortunately, at the pH at which this species forms, the binary CuH(Tyr)_2_ form is also observed at a significant concentration, which prevents a spectroscopic analysis. Therefore, the mode of coordination was determined through quantum-chemical calculations (see Section 2.3.4).

### 2.3. Quantum-Chemical Calculations in Ternary Systems

#### 2.3.1. The Cu(II)/Tyr/Ado System

Preliminary optimization showed that for complexes consisting of one molecule of Tyr as well as one Ado molecule, Cu(Tyr)(Ado)_1 and Cu(Tyr)(Ado)_2 are likely to be observed Appendix A. In Appendix A, the calculated energies, sums of monomers, and interaction energies of possible complexes of Tyr and Ado molecules with copper(II) ions are presented. These results show that the strongest interaction energy among these complexes occurs in the Cu(Tyr)(Ado)_2 complex, with an interaction energy of −251.5 kcal/mol, and tyrosine coordinates with a copper(II) ion via nitrogen and oxygen atoms, while adenosine coordinates only via a nitrogen N(7) atom (simultaneous coordination with nitrogen N(1) and nitrogen N(7) atoms is not possible for steric reasons [44,45]). The optimized structure of this complex is depicted in Figure 12, and the coordinates of both obtained structures are included in Appendix A.

#### 2.3.2. The Cu(II)/Tyr/AMP System

For a complex consisting of monoprotonated or deprotonated Tyr and AMP, only Cu(Tyr)H(AMP)^−^_2 and Cu(Tyr)(AMP)^2−^_2 structures are likely to be observed, while Cu(Tyr)H(AMP)^−^_1 and Cu(Tyr)(AMP)^2−^_1 after optimization are converted into the second structure, respectively (only one scheme of interaction is possible); see Appendix A. The calculated energies, sums of monomers, and interaction energies for these compounds are included in Appendix A. For the complex Cu(Tyr)H(AMP)^−^_2, the interaction energy was −286.2 kcal/mol, and for complex Cu(Tyr)(AMP)^2−^_2, it was −296.9 kcal/mol. In both of these complexes, the tyrosine molecule coordinates with the copper(II) ion via nitrogen and oxygen atoms, while AMP coordinates via one nitrogen atom and one oxygen atom from the phosphate group. The obtained structures are depicted in Figure 13. The coordinates for these structures are included in Appendix A.

#### 2.3.3. The Cu(II)/Tyr/ADP System

For a complex consisting of one molecule of Tyr and one molecule of ADP structures, Cu(Tyr)(ADP)^3−^_1 was not likely to be observed, and after preliminary optimizations, it was converted into Cu(Tyr)(ADP)^3−^_2; see Appendix A. The calculated energies, sums of monomers, and interaction energies of possible complexes of Tyr and ADP molecules with copper(II) ions are presented in Appendix A. The results obtained showed that the strongest interaction energy belonged to the complex Cu(Tyr)(ADP)^3−^_2, which was −310.5 kcal/mol. In this complex, the copper(II) ion was coordinated via nitrogen and oxygen atoms of tyrosine and via a nitrogen atom and two oxygen atoms from both phosphate groups of ADP; see Figure 14. Coordinates for those structures are included in Appendix A.

#### 2.3.4. The Cu(II)/Tyr/ATP System

For a complex consisting of one molecule of H(Tyr) and one molecule of ATP with a copper(II) ion, only Cu(Tyr)H(ATP)^3−^_1 and Cu(Tyr)H(ATP)^3−^_2 are likely to be observed. The structures Cu(Tyr)H(ATP)^3−^_3 and Cu(Tyr)H(ATP)^3−^_4 after optimizations are converted into the Cu(Tyr)H(ATP)^3−^_2 structure; see Appendix A. In Appendix A, we gathered the calculated energies, the sums of the monomer energies, and the interaction energies of possible complexes of H(Tyr) and ATP molecules with a copper(II) ion. The results obtained showed that the strongest interaction energy belonged to complex Cu(Tyr)H(ATP)^3−^_2, and it was −299.9 kcal/mol. In this complex, tyrosine coordinates with the copper(II) ion via oxygen and nitrogen atoms, while ATP coordinates via a nitrogen atom and the *γ* oxygen atom from the phosphate group. The obtained structure is depicted in Figure 15. Coordinates for those structures are included in Appendix A.

In Appendix A, we gathered the calculated energies, sums of monomers, and interaction energies of possible complexes of Tyr and ATP molecules with a copper(II) ion. The results showed that the strongest interaction energy belonged to the complex Cu(Tyr)(ATP)^4−^_2, and it was −304.0 kcal/mol. In this complex, tyrosine coordinates with the copper(II) ion via oxygen and nitrogen atoms, while ATP coordinates via a nitrogen atom and oxygen atoms from the *γ*-phosphate group. The obtained structure is depicted in Figure 16 below. Coordinates for all obtained structures are included in Appendix A.

## 3. Materials and Methods

### 3.1. Chemicals

L-Tyrosine (Tyr), C_9_H_11_NO_3_ (purity 98%); adenosine (Ado), C_10_H_13_N_5_O_4_ (purity 99%); adenosine 5′-monophosphate monohydrate (AMP), C_10_H_14_N_5_O_7_P × H_2_O (purity 97%), adenosine 5’-diphosphate sodium salt (ADP), C_10_H_14_N_5_O_10_P_2_Na (purity 97%); adenosine-5’-triphosphate disodium salt trihydrate (ATP), C_10_H_14_N_5_O_13_P_3_Na_2_ × 3H_2_O (purity 99%); and copper(II) nitrate(V) trihydrate, Cu(NO_3_)_2_ × 3H_2_O (purity 99%), were purchased from Sigma-Aldrich and were used without further purification. The concentrations of copper(II), Cl^−^, as well as Na^+^ ions were determined through the method of inductively coupled plasma optical emission spectrometry (ICP OES). All solutions and experiments were prepared using demineralized, carbonate-free water. D_2_O, NaOD, DCl, and diethyl ether were purchased from Sigma-Aldrich.

### 3.2. Potentiometric Measurements

Potentiometric titrations were conducted using a Metrohm 702 SM Titrino system with an autoburette and a combined glass electrode - Metrohm 6.0233.100 (Metrohm AG, Herisau, Switzerland), calibrated against standard phthalate (pH = 4.002) and borax (pH = 9.225) buffers obtained from Merck (Darmstadt, Germany), according to the hydrogen ion concentration [63]. Titrations were carried out under an argon (5N) atmosphere, at a constant ionic strength (KNO_3_, μ = 0.1 M), and under a controlled temperature of 20 ± 1 °C. The pH range was 2.5–11.0. A CO_2_-free NaOH solution (~0.20 M) was used as the titrant, and the initial sample volume was 30 mL. Binary systems were studied at Cu(II):ligand ratios of 1:1, 1:2, and 1:4, while in ternary systems, M:L:L′ ratios of 1:1:1 and 1:2:2 were used (where L = Tyr, L′ = Ado, AMP, ADP, or ATP). The HYPERQUAD 2008 program [63] was used to determine the complex stability constants (log*β*). For each system, at least six titrations were performed, and approximately 250 data points per titration curve were included in the model-fitting process. The charges of the forming complexes were omitted for clarity.

### 3.3. Spectroscopic Measurements

The mode of coordination was established based on spectroscopic studies. The spectra were recorded at the pH where the concentration of the complex forms was the highest, based on equilibrium studies.

#### 3.3.1. VIS Spectroscopy

Samples for the visible spectroscopy studies were prepared in H_2_O, and the metal concentration was the same as that used in the potentiometric titrations (1 × 10^−3^ M) at the ratios M:L = 1:2 and M:L:L’= 1:1:1. The spectra were recorded at room temperature on an Evolution 300 UV/VIS Thermo Fisher Scientific spectrophotometer (Thermo Electron Scientific Instruments LLC, Madison, WI, USA) using a quartz glass cuvette with a 1 cm path length.

#### 3.3.2. EPR Spectroscopy

The EPR studies were carried out at −196 °C using glass capillary tubes (volume = 130 μm^3^). The concentration of Cu(II) was 1 × 10^−3^ M in a water:ethylene glycol mixture (3:1), and the metal:ligand ratios were 1:2 and 1:2:2 for binary and ternary systems, respectively. The spectra were recorded on an SE/X 2547 Radiopan spectrometer (Radiopan, Poznan, Poland). All experimental spectra were simulated using the computer program EPRsim32 [64]

#### 3.3.3. Infrared Spectroscopy (IR)

IR spectra were recorded on an IR Spirit Fourier Transform Infrared Spectrophotometer (Shimadzu, Kyoto, Japan). Due to the low solubility of tyrosine, the measurements were carried out only for solid free tyrosine and the [CuH_2_(Tyr)_2_(H_2_O)] × 1.5H_2_O complex.

### 3.4. The Crystal Structure

Crystals of [CuH_2_(Tyr)_2_(H_2_O)] × 1.5H_2_O were obtained after 5 days via slow diffusion of Et_2_O into an aqueous solution of the sample at a pH = 7.8 and were prepared at the same concentrations as those for the potentiometric titrations. The reflection intensities for [CuH_2_(Tyr)_2_(H_2_O)] × 1.5H_2_O were measured on an Oxford Diffraction SuperNova Atlas diffractometer equipped with a Cu *Kα* radiation source (1.54184 Å) at room temperature. Data collection, data reduction, and analytical absorption correction were performed using CrysAlisPRO software, version 1.171.43.141a [65]. Olex2 was used as an interface for the structure solution and refinement [66]. The crystal structure of [CuH_2_(Tyr)_2_(H_2_O)] × 1.5H_2_O was solved through the intrinsic phasing method using ShelXT [67] and refined using full-matrix least-squares on *F*^2^ using ShelXL [68]. Hydrogen atoms were positioned geometrically and refined using a riding model. The solvent mask tool in Olex2 was applied due to the inability to model disordered water molecules satisfactorily [66]. The electron count was determined to be 32 $\overset{-}{e}$/cell in an accessible void volume of 157 Å^3^, which indicated the presence of three water molecules in the unit cell.

### 3.5. The Quantum-Chemical Calculations

The interaction energy between tyrosine or monoprotonated tyrosine and copper (II) ions in an aqueous solution was calculated for three possible schemes of interaction for complexes consisting of one molecule of monoprotonated tyrosine with a copper(II) ion (denoted as Cu_H_(Tyr)1, Cu_H(Tyr)2, and Cu_H(Tyr)3; see Appendix A); three possible schemes of interaction for complexes consisting of two molecules of protonated tyrosine with a copper(II) ion (denoted as Cu_H_2_(Tyr)_2__1, Cu_H_2__(Tyr)_2__2, and Cu_H_2__(Tyr)_2__3; see Appendix A); and three possible schemes of interaction for complexes consisting of one monoprotonated and one deprotonated tyrosine molecule with a copper(II) ion (denoted as Cu_H_(Tyr)_2__1, Cu_H_(Tyr)_2__2, and Cu_H_(Tyr)_2__3; see Appendix A). We also made calculations for two possible schemes of interaction of complexes consisting of one molecule of deprotonated tyrosine and one molecule of adenosine with a copper (II) ion (denoted as Cu(Tyr)(Ado)_1 and Cu(Tyr)(Ado)_2; see Appendix A). We also made calculations for complexes consisting of one molecule of monoprotonated tyrosine and one molecule of deprotonated adenosine-5’-monophosphate and for one molecule of deprotonated tyrosine and one molecule of deprotonated adenosine-5’-monophosphate with a copper(II) ion in two interaction schemes for each complex (denoted as Cu(Tyr)H(AMP)^−^_1, Cu(Tyr)H(AMP)^−^_2 and Cu(Tyr)H(AMP)^2−^_1, Cu(Tyr)H(AMP)^2−^_2, respectively; see Appendix A). We also investigated complexes consisting of one deprotonated tyrosine molecule and one molecule of adenosine-5’-diphosphate with a copper(II) ion in three possible interaction schemes: Cu(Tyr)(ADP)^3−^_1, Cu(Tyr)(ADP)^3−^_2, and Cu(Tyr)(ADP)^3−^_3 (see Figure 11). For complexes consisting of one molecule of monoprotonated tyrosine and one molecule of deprotonated adenosine-5’-triphosphate with a copper(II) ion and one molecule of deprotonated tyrosine and one molecule of deprotonated adenosine-5’-triphosphate with a copper(II) ion, we made calculations for four possible interaction schemes (the first denoted as Cu(Tyr)H(ATP)^3−^_1, Cu(Tyr)H(ATP)^3−^_2, Cu(Tyr)H(ATP)^3−^_3, and Cu(Tyr)H(ATP)^3−^_4 and the second denoted as Cu(Tyr)(ATP)^4−^_1, Cu(Tyr)(ATP)^4−^_2, Cu(Tyr)(ATP)^4−^_3, and Cu(Tyr)(ATP)^4−^_4; see Appendix A). The energy calculations were performed using the M06 [69] method and the SDD [70] basis set, which are recommended for non-covalent interactions [33,71,72,73,74,75]. To account for the effect of an aqueous solvent, the Polarizable Continuum Model (PCM) was employed [76]. A frequency analysis [77] was conducted to assess the thermochemistry data. All calculations were performed in Gaussian 16 [78].

## 4. Conclusions

In this work, we have attempted to determine the mode of interaction in complexes formed in a Cu(II)/Tyr binary system, as well as in Cu(II)/Tyr/Ado (AMP, ADP, or ATP) ternary systems. It was also important to establish how the number of phosphate groups in the chain influences the mode of coordination of the nucleotides in the ternary species.

In the binary Cu(II)/Tyr system, we found the formation of four complexes, i.e., CuH_2_(Tyr), CuH(Tyr), CuH_2_(Tyr)_2_, and CuH(Tyr)_2_. With an increasing pH, the mode of coordination for these species changes from the {O} chromophore through {1N,1O} to the {2N,2O} type. Moreover, a solid CuH_2_(Tyr)_2_ complex was obtained, for which the formula [CuH_2_(Tyr)_2_(H_2_O)] × 1.5H_2_O was determined. For this species, both in solution as well as in crystal form, the same mode of coordination was determined.

When a second ligand, i.e., Ado, AMP, ADP, or ATP, is introduced into the Cu(II)/Tyr binary system, MLH_x_L’ and mixed MLL’ protonated complexes are formed. Only in the ATP system were no MLL’(OH)_x_ species found. In all systems, ternary forms were formed over the entire pH range studied, and the anchoring complexes were binary species of copper(II) ions with tyrosine.

In the Cu(II)/Tyr/Ado system, up to a pH of ~7.0, in the interaction with the metal ions, an oxygen atom from the carboxyl group –COO^−^ of Tyr as well as a nitrogen atom from the purine ring of the nucleoside are involved. Above a physiological pH, the nitrogen atom from the -NH_2_ group of the amino acid molecule also participates in the coordination. In the corresponding system with AMP, above a pH = 7.0, the oxygen atom from the phosphate group of this ligand is also involved in the interaction. Moreover, an increase in the number of phosphate groups in the ADP and ATP molecules has no effect on their participation in the coordination of ternary species, and these ligands interact just like in binary species (i.e., in ADP, both *α* and *β*-phosphate groups [44], and in ATP, only the *γ*-phosphate group [50]). Furthermore, it was observed that the introduction of a second ligand into the Cu(II)/Tyr system did not change the tyrosine coordination mode over the entire pH range studied.

For the chosen complexes formed in the investigated systems, quantum-chemical calculations were performed, and on the basis of the results obtained, the most energetically favorable modes of interaction were determined.

## Figures and Tables

**Figure 1 ijms-26-08865-f001:**
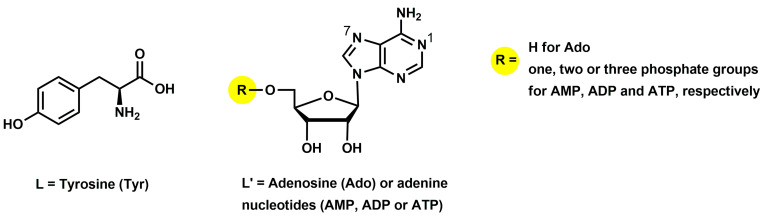
The chemical formulae of the studied ligands.

**Figure 2 ijms-26-08865-f002:**
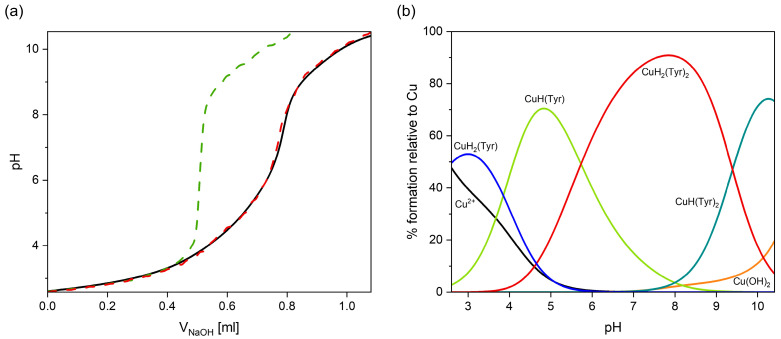
(**a**) Experimental and simulated titration curves for the Cu(II)/Tyr system: green—experimental curve (without complex formation); red—experimental curve; black—simulated curve (with complex formation); (**b**) distribution diagram for the Cu(II)/Tyr system, C_Cu_^2+^ = 1 × 10^−3^ M, C_Tyr_ = 2 × 10^−3^ M.

**Figure 3 ijms-26-08865-f003:**
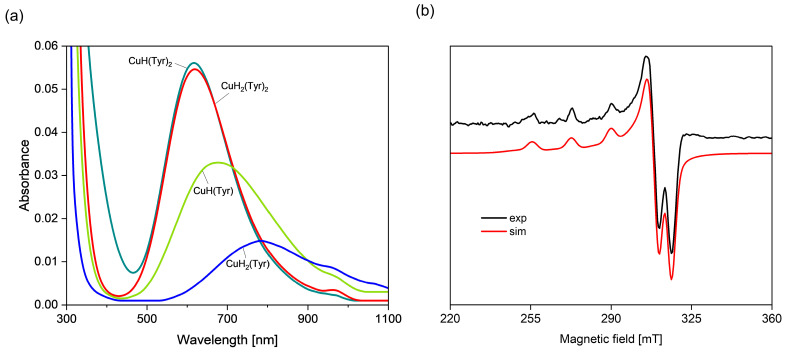
(**a**) The VIS spectra of the Cu(II)/Tyr system; (**b**) experimental and simulated EPR spectra for the Cu(II)/Tyr system at a pH = 10.2; C_Cu_^2+^ = 1 × 10^−3^ M, C_Tyr_ = 2 × 10^−3^ M.

**Figure 4 ijms-26-08865-f004:**
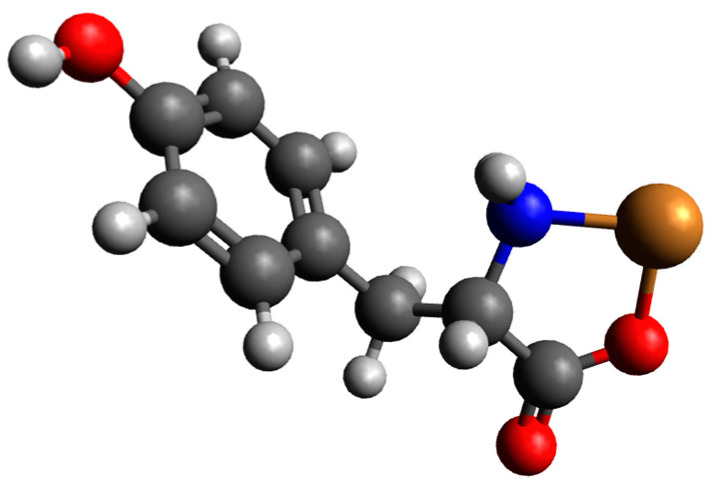
The optimized structure of a complex of monoprotonated tyrosine with a copper(II) ion; trongest interaction between the two molecules of dopamine and the copper(II) ion (Cu_DA2_3); dark grey—carbon atoms; light grey—hydrogen atoms; red—oxygen atoms; blue—nitrogen atom and brown—copper(II) ion.

**Figure 5 ijms-26-08865-f005:**
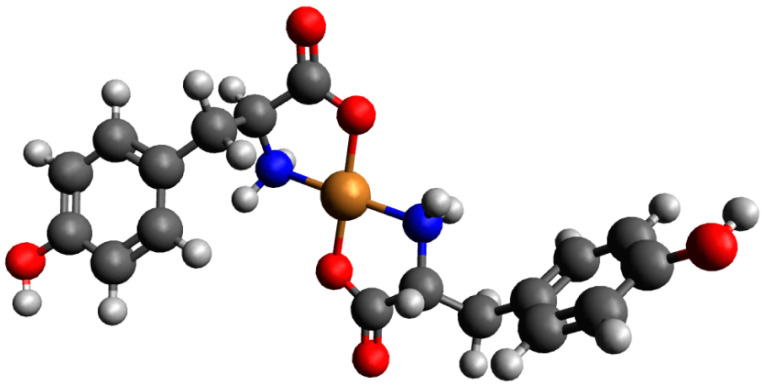
The structure of the complex with the strongest interaction between two monoprotonated tyrosine molecules and a copper(II) ion (Cu_H_2__(Tyr)_2__2); dark grey—carbon atoms; light grey—hydrogen atoms; red—oxygen atoms; blue—nitrogen atoms and brown—copper(II) ion.

**Figure 6 ijms-26-08865-f006:**
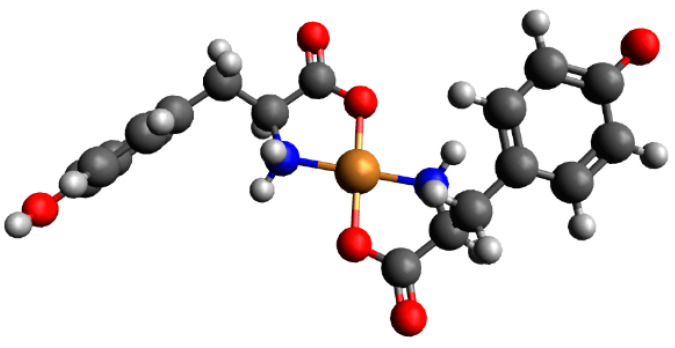
The structure of the complex with the strongest interaction between one monoprotonated and one deprotonated tyrosine molecules and a copper(II) ion (Cu_H_(Tyr)_2__1); dark grey—carbon atoms; light grey—hydrogen atoms; red—oxygen atoms; blue—nitrogen atoms and brown—copper(II) ion.

**Figure 7 ijms-26-08865-f007:**
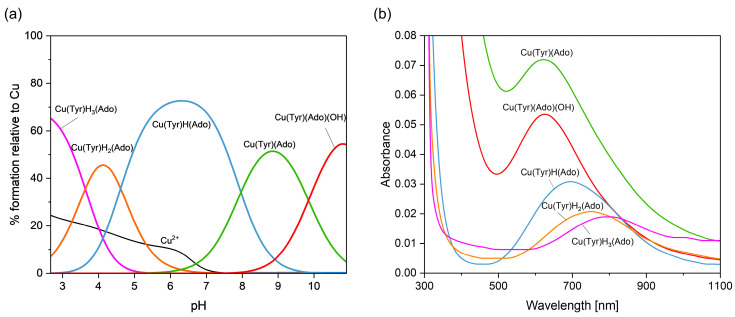
(**a**) The distribution diagram for the Cu(II)/Tyr/Ado system; the percentage of the species refers to total Cu(II); (**b**) the VIS spectra of the Cu(II)/Tyr/Ado system; C_Cu_^2+^ _=L = L’_ = 1 × 10^−3^ M.

**Figure 8 ijms-26-08865-f008:**
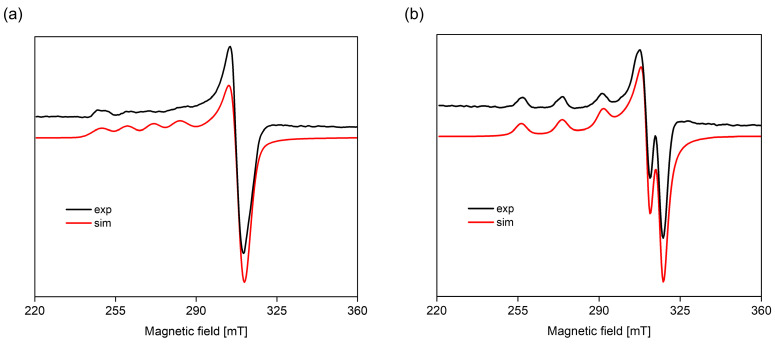
The experimental and simulated EPR spectra for the Cu(II)/Tyr system (**a**) at a pH = 4.1 and (**b**) at a pH = 10.5; C_Cu_^2 +^ _=L = L’_ = 1 × 10^−3^ M.

**Figure 9 ijms-26-08865-f009:**
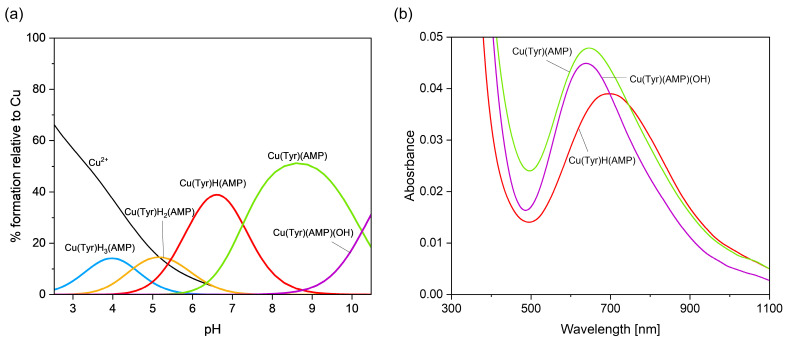
(**a**) The distribution diagram for the Cu(II)/Tyr/AMP system; the percentage of the species refers to the total Cu(II); (**b**) the VIS spectra of the Cu(II)/Tyr/AMP system; C_Cu_^2 +^ _=L = L’_ = 1 × 10^−3^ M.

**Figure 10 ijms-26-08865-f010:**
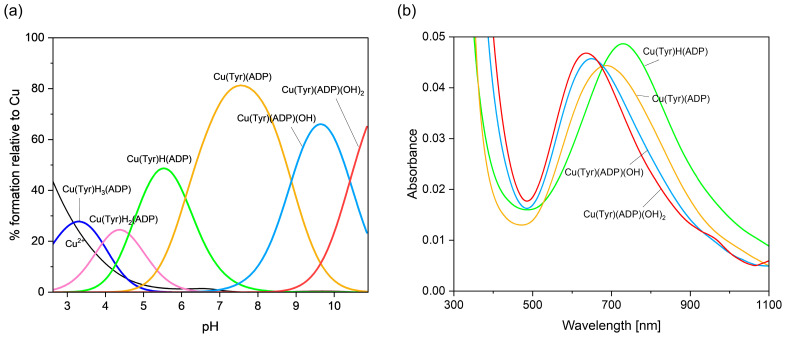
(**a**) The distribution diagram for the Cu(II)/Tyr/ADP system; the percentage of the species refers to the total Cu(II); (**b**) the VIS spectra of the Cu(II)/Tyr/ADP system; C_Cu_^2+^ = 1 × 10^−3^ M, C_L = L’_ = 1 × 10^−3^ M.

**Figure 11 ijms-26-08865-f011:**
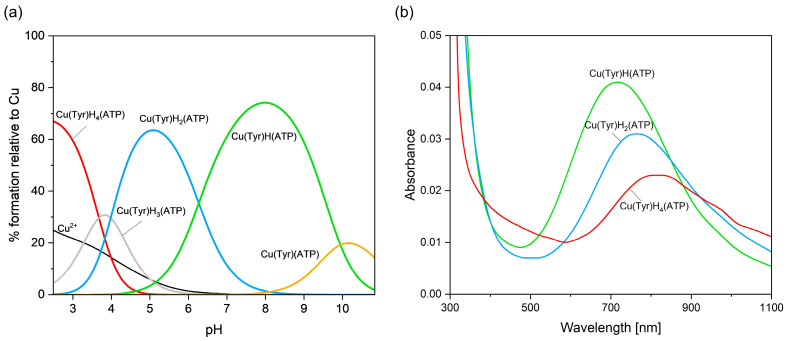
(**a**) A distribution diagram for the Cu(II)/Tyr/ATP system: the percentage of the species refers to the total Cu(II); (**b**) the VIS spectra of the Cu(II)/Tyr/ATP system; C_Cu_^2+^ _=L = L’_ = 1 × 10^−3^ M.

**Figure 12 ijms-26-08865-f012:**
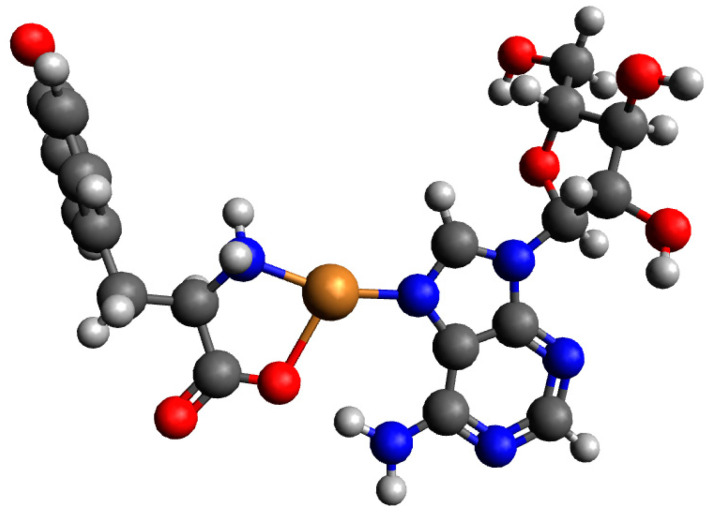
The structure of the complex with the strongest interaction between the deprotonated tyrosine molecule, the adenosine molecule, and the copper(II) ion: Cu(Tyr)(Ado)_2; dark grey—carbon atoms; light grey—hydrogen atoms; red—oxygen atoms; blue—nitrogen atoms and brown—copper(II) ion.

**Figure 13 ijms-26-08865-f013:**
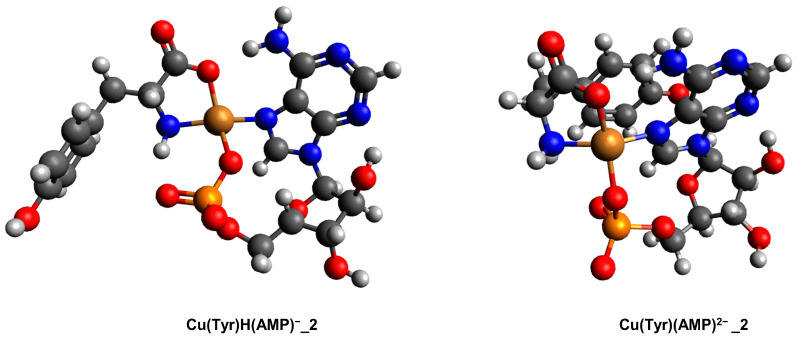
Structures of the complexes of one molecule of monoprotonated/deprotonated tyrosine and one molecule of AMP with a copper(II) ion: (Cu(Tyr)H(AMP)^−^_2 and Cu(Tyr)(AMP)^2−^_2; dark grey—carbon atoms; light grey—hydrogen atoms; red—oxygen atoms; blue—nitrogen atoms, orange—phosphorous atoms and brown—copper(II) ions.

**Figure 14 ijms-26-08865-f014:**
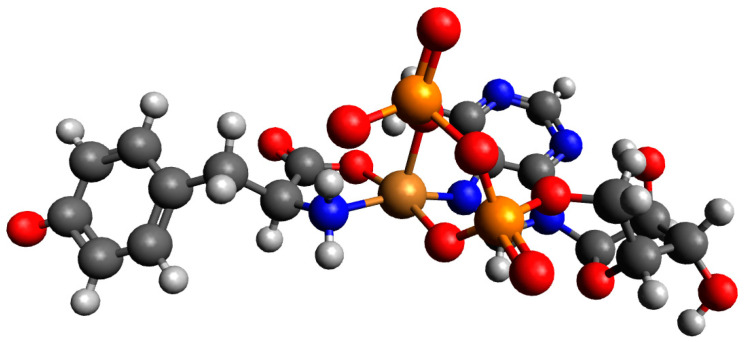
The structure of the complex with the strongest interaction between one molecule of deprotonated tyrosine and one molecule of ADP and the copper(II) ion (Cu(Tyr)(ADP)^3−^_2); dark grey—carbon atoms; light grey—hydrogen atoms; red—oxygen atoms; blue—nitrogen atoms, orange—phosphorous atoms and brown—copper(II) ions.

**Figure 15 ijms-26-08865-f015:**
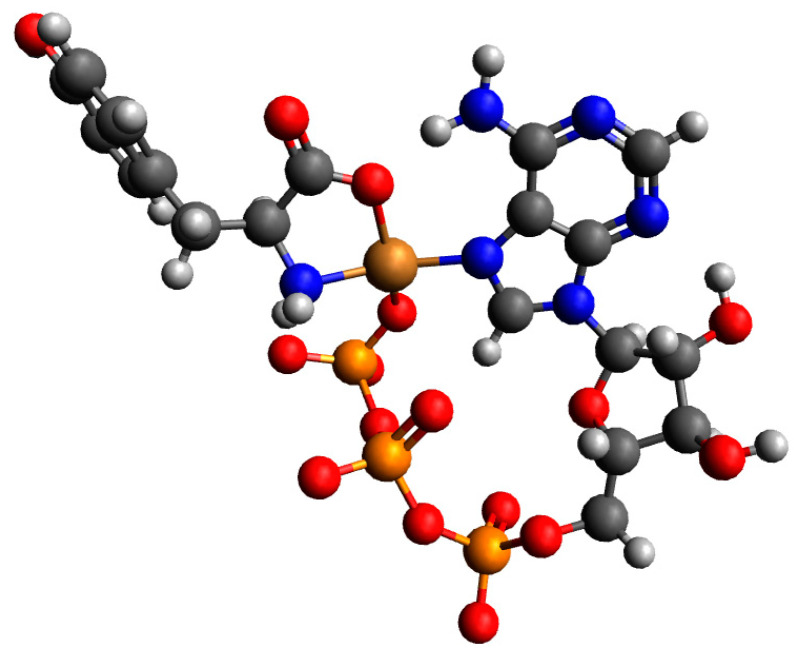
The structure of the complex with the strongest interaction between one molecule of monoprotonated tyrosine, one molecule of ATP, and a copper(II) ion: Cu(Tyr)H(ATP)^3−^_2; dark grey—carbon atoms; light grey—hydrogen atoms; red—oxygen atoms; blue—nitrogen atoms, orange—phosphorous atoms and brown—copper(II) ions.

**Figure 16 ijms-26-08865-f016:**
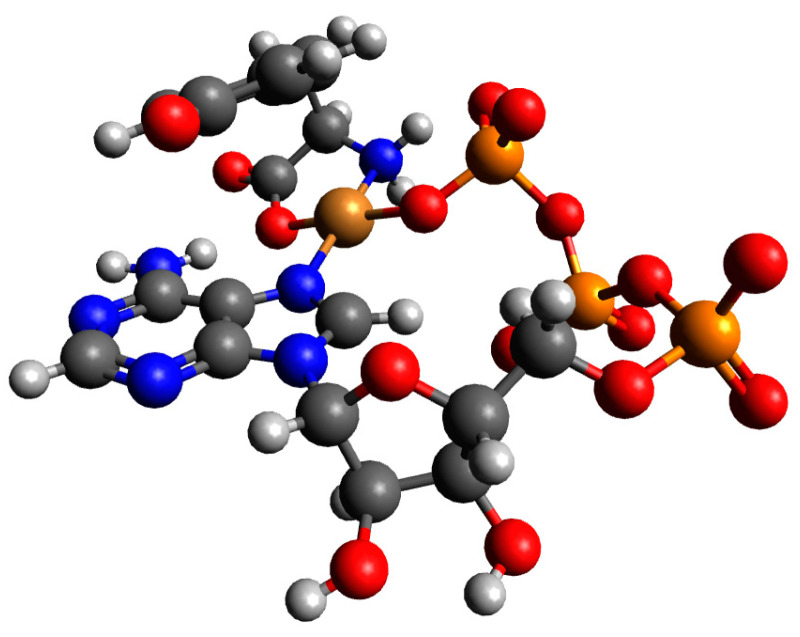
The structure of the complex with the strongest interaction between one molecule of deprotonated tyrosine, one molecule of ATP, and a copper(II) ion: Cu(Tyr)(ATP)^4−^_2; dark grey—carbon atoms; light grey—hydrogen atoms; red—oxygen atoms; blue—nitrogen atoms, orange—phosphorous atoms and brown—copper(II) ions.

**Table 1 ijms-26-08865-t001:** The protonation constants of tyrosine, overall stability constants (log*β*), and equilibrium constants (log*K_e_*) for complexes formed in the binary Cu(II)/Tyr system.

System	Species	Counterions	log*β*	log*K_e_*	Ref.
Tyr	H_3_(Tyr)^+^	OH^−^	21.81 ± 0.03	2.52	21.61 [39], 22.48 [40], 21.47 [38]
	H_2_(Tyr)	-	19.29 ± 0.02	9.01	19.18 [39], 19.47 [40], 19.27 [38], 19.66 [41]
	H(Tyr)^−^	H^+^	10.28 ± 0.02	10.28	10.16 [39], 10.12 [40], 10.22 [38], 10.47 [41]
Cu(II)/Tyr	CuH_2_(Tyr)^2+^	2NO_3_^−^	22.39 ± 0.05	3.10	-
	CuH(Tyr)^+^	NO_3_^−^	18.54 ± 0.03	8.26	18.28 [41], 17.79 [42], 19.34 [43]
	CuH_2_(Tyr)_2_	-	35.38 ± 0.03	6.56	35.68 [41], 34.64 [42], 36.41 [43]
	CuH(Tyr)_2_^2−^	2H^+^	25.99 ± 0.03	7.45	25.32 [42], 25.47 [51]

**Table 2 ijms-26-08865-t002:** The visible and EPR spectroscopic data and chromophore types for the Cu(II)/Tyr complexes.

Species	pH	λ_max_ [nm]	g_||_	g_┴_	A_||_ [10^−4^ cm^−1^]	A_┴_ [10^−4^ cm^−1^]	Chromophore
CuH_2_(Tyr)	3.0	780	2.349	2.066	140	10	{1O}
CuH(Tyr)	4.8	670	2.307	2.059	174	13	{1N,1O}
CuH_2_(Tyr)_2_	7.8	620	2.257	2.054	181	10	{2N,2O}
CuH(Tyr)_2_	10.2	620	2.257	2.052	184	10	{2N,2O}

**Table 3 ijms-26-08865-t003:** Overall stability constants (log*β*) of complexes formed in Cu(II)/Tyr/Ado, Cu(II)/Tyr/AMP, Cu(II)/Tyr/ADP, and Cu(II)/Tyr/ATP ternary systems.

Systems	Species	Counterions	log*β*
Cu(II)/Tyr/Ado	Cu(Tyr)H_3_(Ado)^3+^	3NO_3_^−^	31.40 ± 0.07
	Cu(Tyr)H_2_(Ado)^2+^	2NO_3_^−^	27.77 ± 0.06
	Cu(Tyr)H(Ado)^+^	NO_3_^−^	23.15 ± 0.06
	Cu(Tyr)(Ado)	-	15.19 ± 0.08
	Cu(Tyr)(Ado)(OH)^−^	H^+^	5.35 ± 0.08
Cu(II)/Tyr/AMP	Cu(Tyr)H_3_(AMP)^+^	NO_3_^−^	32.11 ± 0.07
	Cu(Tyr)H_2_(AMP)	-	27.59 ± 0.07
	Cu(Tyr)H(AMP)^−^	H^+^	22.14 ± 0.03
	Cu(Tyr)(AMP)^2−^	2H^+^	14.87 ± 0.04
	Cu(Tyr)(AMP)(OH)^3−^	3H^+^	4.62 ± 0.05
Cu(II)/Tyr/ADP	Cu(Tyr)H_3_(ADP)	-	33.65 ± 0.03
	Cu(Tyr)H_2_(ADP)^−^	H^+^	29.73 ± 0.03
	Cu(Tyr)H(ADP)^2−^	2H^+^	25.11 ± 0.02
	Cu(Tyr)(ADP)^3−^	3H^+^	19.01 ± 0.02
	Cu(Tyr)(ADP)(OH)^4−^	4H^+^	10.14 ± 0.02
	Cu(Tyr)(ADP)(OH)_2_^5−^	5H^+^	−0.27 ± 0.02
Cu(II)/Tyr/ATP	Cu(Tyr)H_4_(ATP)	-	38.44 ± 0.04
	Cu(Tyr)H_3_(ATP)^−^	H^+^	34.76 ± 0.04
	Cu(Tyr)H_2_(ATP)^2−^	2H^+^	30.78 ± 0.04
	Cu(Tyr)H(ATP)^3−^	3H^+^	24.51 ± 0.02
	Cu(Tyr)(ATP)^4−^	4H^+^	14.56 ± 0.07

log*β* for: Cu(OH)_2_ = −13.13 [57]; H(Ado)^+^ = 3.92 ± 0.01; Cu(Ado)^2+^ = 2.88 ± 0.15; Cu(Ado)(OH)_2_ = −11.41 ± 0.08; Cu(Ado)(OH)_3_^−^ = −20.92 ± 0.14 [48]; H_2_(AMP) = 10.45 ± 0.02; H(AMP)^−^ = 6.43 ± 0.02; Cu(AMP) = 3.02 ± 0.08; Cu(AMP)(OH)^−^ = −3.82 ± 0.05 [49]; H_2_(ADP)^−^ = 10.63 ± 0.02; H(ADP)^2−^ = 6.55 ± 0.01; CuH(ADP) = 10.84 ± 0.03; Cu(ADP)^−^ = 6.99 ± 0.06; Cu(ADP)(OH)^2−^ = −1.03 ± 0.02 [44]; H_2_(ATP)^2−^ = 10.88 ± 0.02; H(ATP)^3−^ = 6.50 ± 0.01; CuH(ATP)^−^ = 10.53 ± 0.03; Cu(ATP)^2−^ = 6.63 ± 0.02; Cu(ATP)(OH)^3−^ = −1.25 ± 0.02 [50].

**Table 4 ijms-26-08865-t004:** VIS and EPR spectroscopic parameters for complexes formed in Cu(II)/Tyr/Ado, Cu(II)/Tyr/AMP, Cu(II)/Tyr/ADP, and Cu(II)/Tyr/ATP ternary systems.

Systems	Species	pH	λ_max_ [nm]	g_||_	g_┴_	A_||_ [10^−4^ cm^−1^]	A_┴_ [10^−4^ cm^−1^]
Cu(II)/Tyr/Ado	Cu(Tyr)H_3_(Ado)	3.0	790	2.402	2.069	130	5
	Cu(Tyr)H_2_(Ado)	4.1	750	2.370	2.066	120	8
	Cu(Tyr)H(Ado)	6.2	700	2.301	2.069	174	8
	Cu(Tyr)(Ado)	8.8	635	2.251	2.055	185	5
	Cu(Tyr)(Ado)(OH)	10.5	625	2.250	2.051	190	4
Cu(II)/Tyr/AMP	Cu(Tyr)H(AMP)	6.6	670	2.301	2.059	180	11
	Cu(Tyr)(AMP)	8.6	650	2.300	2.063	174	5
	Cu(Tyr)(AMP)(OH)	10.7	635	2.258	2.057	190	7
Cu(II)/Tyr/ADP	Cu(Tyr)H(ADP)	5.6	725	2.362	2.068	138	13
	Cu(Tyr)(ADP)	7.5	675	2.325	2.067	146	11
	Cu(Tyr)(ADP)(OH)	9.6	650	2.322	2.038	124	22
	Cu(Tyr)(ADP)(OH)_2_	10.5	640	2.256	2.053	186	9
Cu(II)/Tyr/ATP	Cu(Tyr)H_4_(ATP)	2.5	800	2.397	2.072	129	4
	Cu(Tyr)H_2_(ATP)	5.2	760	2.363	2.066	147	8
	Cu(Tyr)H(ATP)	8.0	705	2.316	2.047	155	6

## Data Availability

The data are contained within the article and Appendix A.

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
