# Peer review of "Characterization of the Binding Modes of Cu2+ Ions with Tyrosine and Ado, AMP, ADP, and ATP: A Comprehensive Potentiometric, Spectroscopic, and Computational Approach"

_ijms, 2025, doi:10.3390/ijms26188865_

Round 1

Reviewer 1 Report

Comments and Suggestions for Authors

The review report is provided in the attached file.

Author Response

Author response to reviewer 1:

Summary

The study investigates the interaction of copper(II) ions with tyrosine and nucleotides (adenosine, AMP, ADP, and ATP) using potentiometric methods, spectroscopic techniques, and DFT calculations. In the binary systems, tyrosine coordinates mainly through its amino and carboxylate groups, whereas in the ternary systems the nucleotides provide additional coordination modes—particularly via their phosphate groups—without displacing the central role of tyrosine. Overall, the findings highlight a cooperative contribution of both ligands in stabilizing copper complexes, offering valuable insight into the regulation of such interactions in biologically relevant systems and presenting an integrated experimental–theoretical perspective.

Revisions

The study is interesting and demonstrates strong theoretical potential, being well-structured and carefully analyzed. However, it requires certain adjustments and additional analyses to confirm some of the proposed interactions from a theoretical standpoint, as well as complementary calculations to strengthen the overall findings.

The use of the M06/SDD level of theory is reasonable; however, for a study that relies on such fine comparisons of energy, a more suitable choice would be B97M-V in combination with the def2-TZVP basis set (employing the def2-ECP effective core potential for copper). B97M-V belongs to the family of modern meta-GGA functionals and has demonstrated outstanding performance across a wide range of benchmark studies—not only in thermochemistry and bond energies but also in the accurate description of noncovalent interactions that are particularly relevant in biological systems, such as hydrogen bonding and aromatic stacking. Unlike older functionals such as M06, which tend to overestimate dispersion, B97M-V explicitly incorporates nonlocal correlation through the VV10 term, allowing for a more balanced representation of dispersion forces without relying on external empirical corrections.

With regard to the basis set, the def2 family represents the current standard in computational chemistry due to its systematic construction and well-balanced design. In particular, def2-TZVP provides a triple-ζ quality with sufficient polarization functions to accurately describe both the metal–ligand bonds and the secondary interactions surrounding the complex. For copper, the use of the relativistic def2-ECP pseudopotential effectively reduces computational cost without compromising the accuracy of the valence electron treatment, which is especially relevant for large bioinorganic complexes.

In reproducing some of your calculations with B97M-V/def2-TZVP (with ECP for Cu), I observed that the interaction energies are systematically less negative than those obtained with M06/SDD, which is consistent with the well-known tendency of M06 to overestimate dispersion. Despite this reduction, the relative ordering of isomers and the coordination patterns inferred from pH-dependent speciation and spectroscopic data are preserved. In the ternary complexes with nucleotides, B97M-V produces a smaller perturbation of the weak contacts, leading to a slightly lower energetic gain upon incorporation of the base/phosphate; nevertheless, the overall conclusions regarding which groups enter the first coordination sphere (ADP: α/β; ATP: γ) remain unaffected. While I understand that the main study was carried out using M06/SDD, it would be valuable to include in the supplementary material at least one of the key complexes recalculated with B97M-V/def2-TZVP, in order to illustrate the differences and allow the manuscript to explicitly acknowledge that, although M06/SDD is reasonable, the latter method is more appropriate for fine energetic comparisons.

Response:

We would like to thank anonymous Reviewer for the detailed recommendation and for reproducing some of our calculations using B97M-V/def2-TZVP (with ECP for Cu). We acknowledge the advantages of B97M-V for balanced representation of dispersion forces and noncovalent interactions, as well as the suitability of the def2-TZVP basis set for bioinorganic complexes. The Reviewer's observations that the relative ordering of isomers and coordination patterns are preserved, despite systematically less negative interaction energies, align with our results. Due to insufficient time given to make corrections we are unable to recalculate all of investigated compounds. We are very thankful for suggested methods and basis sets which we will certainly use in our future works.

In addition to reporting interaction energies, the authors should also calculate and present the standard Gibbs free energy of binding (ΔG°, including thermal and standard-state corrections). This would allow a more direct comparison with the potentiometrically determined logβ values and provide further insight into complex stability. The calculation is straightforward: one simply takes the optimized structure and performs a single-point calculation with the same functional and basis set, including energy and frequency (without re-optimizing the geometry). This yields the free energy of each species, from which the corresponding ΔG° can be readily obtained.

Response:

We would like to thank anonymous Reviewer for this suggestion. We fully agree to calculate and present ΔG° (including thermal and standard-state corrections). As recommended, we made it with use our existing optimized structures to perform single-point frequency calculations with the M06/SDD level of theory. The resulting ΔG° values are now included in the Supplementary Material.

The authors refer to interaction regions between the molecules, i.e., reactive zones. It is therefore highly recommended that they compute and plot the dual descriptor of reactivity, as this would substantially strengthen the study without requiring significant additional effort. The Multiwfn 3.8 program is freely available and provides hundreds of analyses directly from the Gaussian output files, without the need to rerun the calculations. Generating the dual descriptor plots typically takes only a few minutes, yet it is a very powerful tool. For illustration, Figure 1R shows an example of a molecule analyzed with the dual descriptor, clearly highlighting its nucleophilic and electrophilic regions.

Figure 1R: Example of a dual reactivity descriptor

The dual descriptor of reactivity can be obtained graphically either with Multiwfn or directly with Gaussian. If the authors prefer the latter, it requires more steps, since by definition the dual descriptor corresponds to the electron density of the LUMO minus that of the HOMO. A reasonable approximation is obtained by squaring the HOMO and LUMO orbitals to derive their respective electron densities, which can then be subtracted. However, this procedure is rather cumbersome, and in my experience Gaussian occasionally presents difficulties when visualizing the dual descriptor. For this reason, I strongly recommend using Multiwfn for this purpose, as it is more straightforward and reliable.

The investigation of copper complexes with protonated ligands is certainly interesting, since copper can either coordinate directly to the protonated ligands, as shown by the authors, or alternatively displace the proton to form the complex. It is therefore crucial to establish whether the proton is actually displaced, and what type of bond is formed with the metal atom. For instance, in Figure 5 the authors depict a copper center bound to a nitrogen atom carrying two protons (a positively charged site), which is highly unstable due to the strong repulsion between two electrophilic centers. Although Figure S1 in the supplementary material suggests the coexistence of this species, this makes it even more necessary to determine its free energy and to conduct a kinetic study to evaluate the activation energy of the complexes, for example through IRC or OPDOS analyses, which I will comment on later. An IRC profile, in particular, would allow the reader to visualize the transition state, the overall reaction coordinate, and the possible existence of intermediates. As illustrated by Figure 12, it is not sufficient to compute only the Gibbs free energy of formation; it is also important to examine the activation barrier and to clarify whether the reported structure corresponds to a true product or rather to a transition state leading to an intermediate, which may then evolve through a second transition state into the final complex. The strong interactions reported in the manuscript do not necessarily prove that the species is a stable product; in some cases, it could instead represent a highly stabilized intermediate along the reaction pathway.

The authors have carried out an impressive and thorough effort in calculating binding energies, and their work is certainly commendable. However, in order to make the nature of the interactions more visible and accessible to the reader, it would be highly advisable to complement these results with a reduced density gradient (RDG) analysis. RDG plots provide qualitative, intuitive visualization of the intensity and character of interatomic interactions, as illustrated in Figure R2 for a protonated copper complex. Despite the oxygen being in the form of an OH group, the copper center is still capable of establishing a strong interaction with it. This interaction induces a 180° reorientation of the proton due to repulsion from the copper atom, eventually leading to its departure as a free proton.

Figure R2. Example of interaction regions shown as RDG surfaces for a reaction intermediate.

This analysis can also be conveniently performed using the freely available program Multiwfn 3.8. The software offers several types of RDG visualizations, allowing the authors to focus on specific interactions of interest. It also enables the detection of extremely weak interactions, as well as electronic repulsions, which would further strengthen the interpretation of the binding motifs discussed in the manuscript.

A critical part of this study is the in-depth quantum analysis of the nature and composition of the bond or interaction through an OPDOS investigation. This type of analysis, which can also be obtained directly from the same computational framework, provides valuable insight into the distribution of electronic density within the interaction. It allows one to assess, in a qualitative manner, whether a bond or interaction is strong or weak, in agreement with the intensity of the RDG plots, where more intense features correspond to stronger interactions. OPDOS analysis yields particularly rich information about the intrinsic character of the bond, indicating, for instance, that if the electronic density lies closer to the Fermi level, the interaction will exhibit a stronger character. Conversely, the presence of electrons in antibonding levels weakens the interaction due to the vacancies created. In such cases, the addition of electrons to these antibonding orbitals can drastically alter both the shape and the nature of the interaction. This is illustrated in Figure R3 for a copper complex.

Figure R3 OPDOS

It is important to pay close attention to the sign convention used in the program, ensuring that the Y-axis is configured so that positive values correspond to bonding interactions and negative values to antibonding ones. It should also be noted that the mere presence of antibonding orbitals does not necessarily imply that the interaction is of poor quality, and this distinction must be carefully considered when interpreting the OPDOS results.

Response:

We would like to thank anonymous Reviewer for his recommendation to perform analyses such as the dual descriptor of reactivity, reduced density gradient (RDG) plots, and overlapped partial density of states (OPDOS) using the Multiwfn 3.8 program, which we recognize as a truly wonderful and powerful tool for providing deep insights into molecular interactions, reactive zones, and electronic properties without necessitating reruns of our DFT calculations. In future we want to explore its capabilities but we must admit that Multiwfn was previously unknown to our team, and given the time constraints for revisions, we are unable to acquire, learn, and implement it proficiently to perform these additional computations and ensure their accurate integration into the manuscript. We have downloaded the Multiwfn program and its tutorial, which we will try to familiarize ourselves with and use in our calculations in the future.

The authors should avoid overloading the main manuscript with excessive numerical data when describing molecular interactions. Instead, they are encouraged to prioritize visual representations that are more accessible to the reader, while leaving most of the numerical details to the Supplementary Material and keeping only the essential values in the main text. Importantly, the reported data should be consistent across the different analyses: for instance, a very weak interaction would typically correspond to a faint RDG isosurface and an OPDOS spectrum with electronic densities shifted away from the Fermi level or dominated by antibonding orbitals. If any of these analyses appear inconsistent, it may indicate an issue with the initial calculations, which also makes these methods a valuable tool for verifying the correctness of the computational setup. Finally, if the authors wish to further quantify the electronic densities contributing to the interactions, they could complement their analysis with a COOP study, which is conceptually related to OPDOS but provides the advantage of evaluating the integrated areas under the curves.

The research team clearly demonstrates strong expertise, both on the experimental and theoretical fronts. That said, I believe the theoretical group could benefit from exploring a broader range of available tools to deepen their analysis. For obvious and ethical reasons, I cannot provide references

to specific works here. It would certainly have been interesting to collaborate directly with the authors, but in my role as referee I must remain impartial and respect the boundaries of this position.

Response:

We would like to thank the Reviewer for the kind words on our expertise and for the ethical note on collaboration boundaries.

Reviewer 2 Report

Comments and Suggestions for Authors

In the reviewed paper, the authors try to determine the modes of coordination of Cu(II)/tyrosine complexes in solutions of different pH values and to study interactions in ternary Cu(II)/tyrosine/adenosine or adenosine phosphate systems. More than twenty complexes are examined in detail in the paper.

My suggestions to improve the article are the following. It would be desirable to present the absorption bands in Figs. 3, 7, 9, 10, and 11 in a wider wavelength range (300 nm–1300 nm) in order to better assess the width of the absorption bands. Also, the authors do not indicate anywhere which quantum chemical package they used to optimize their structures and calculate interaction energies given in the Supplementary Materials.

The spectral shifts with pH changes, which the authors demonstrate in the article (Figs 7,9,10,11), may be partly related to changes in solvation of complexes, and not only to changes in the immediate coordination environment of copper ions. In the approach proposed by the authors, these effects cannot be separated, but this could be well done by calculations using the td-DFT approach.

At lines 158-160 the authors interpret a nature of the d-d band shift observed. Line 161 further states that this is also confirmed by the EPR data from Table 2. Please, explain this idea in more detail, since Table 2 shows that g|| only changes by 0.05, and it is completely unclear how to purposefully link this change to the coordination rearrangement  taking place. A similar question arises also with the Cu(Tyr)H3(Ado) complex, line 271.

Finally, a general request to the authors: please indicate the counterions of the complexes under study where they exist. For example, what is the charge (and counterion) of the complex shown in Fig. 4 if it contains a copper(II) ion?

The article can be published with some minor revisions.

Author Response

Author response to reviewer 2

In the reviewed paper, the authors try to determine the modes of coordination of Cu(II)/tyrosine complexes in solutions of different pH values and to study interactions in ternary Cu(II)/tyrosine/adenosine or adenosine phosphate systems. More than twenty complexes are examined in detail in the paper.

My suggestions to improve the article are the following. It would be desirable to present the absorption bands in Figs. 3, 7, 9, 10, and 11 in a wider wavelength range (300 nm–1300 nm) in order to better assess the width of the absorption bands.

Response:

Thank you for pointing this out. We presented the absorption bands in Figs. 3, 7, 9, 10, and 11 in a wider wavelength range (300 nm–1100 nm). 1100 nm is the wavelength up to which the spectrum can be measured on the spectrophotometer we use. Typical UV-Vis spectrophotometers operate up to around 1100 nm, while measurements above this limit fall into the NIR range and require different instrumentation.

Also, the authors do not indicate anywhere which quantum chemical package they used to optimize their structures and calculate interaction energies given in the Supplementary Materials.

Response:

We would like to thank reviewer for this suggestion. All quantum chemical calculations were made with use of Gaussian 16. We added this information in Materials and Methods section.

The spectral shifts with pH changes, which the authors demonstrate in the article (Figs 7,9,10,11), may be partly related to changes in solvation of complexes, and not only to changes in the immediate coordination environment of copper ions. In the approach proposed by the authors, these effects cannot be separated, but this could be well done by calculations using the td-DFT approach.

Response:

We would like to thank reviewer for this suggestion to employ TD-DFT calculations to separate solvation effects from changes in the coordination environment which is highly valuable. However, within the scope of this study, we prioritized experimental data to directly observe these phenomena under real conditions. Conducting TD-DFT calculations, while potentially informative, would have required additional modeling assumptions and significant computational resources, which were beyond the scope of this work. We believe that our experimental approach provides sufficient evidence for the dominant role of changes in the coordination environment, though we do not rule out the contribution of solvation.

We plan to consider TD-DFT calculations in future studies to further explore these effects, which could complement our current findings. Thank you for highlighting this direction, which we will certainly take into account in our future research.

At lines 158-160 the authors interpret a nature of the d-d band shift observed. Line 161 further states that this is also confirmed by the EPR data from Table 2. Please, explain this idea in more detail, since Table 2 shows that g|| only changes by 0.05, and it is completely unclear how to purposefully link this change to the coordination rearrangement taking place. A similar question arises also with the Cu(Tyr)H3(Ado) complex, line 271.

Response:

We appreciate the reviewer’s request for clarification. In axially elongated Cu(II) complexes with a dx2−y2 ground state, the EPR parameter g∥ and the energy of the d − d transition are both sensitive to the strength and covalency of the equatorial ligand field and to the magnitude of the tetragonal (Jahn–Teller) distortion. According to the standard ligand-field/EPR correlations for Cu(II), strengthening the equatorial field (and/or weakening the axial interaction) increases the d − d transition energy (blue shift) and decreases g∥ (greater equatorial covalency lowers g∥). Thus, the two observables should change in a concerted and internally consistent way.

In our data we indeed observe such concerted changes:

  • CuH(Tyr) → CuH2(Tyr)2: λmax shifts from 670 to 620 nm, while g∥ decreases from 2.307 to 2.257.
  • Cu(Tyr)H3(Ado) → Cu(Tyr)H2(Ado): λmax shifts from 790 to 750 nm, while g∥ decreases from 2.402 to 2.370.

Although a change of ~0.05 in g∥ may appear small in absolute terms, it is substantially larger than the experimental uncertainty for X-band EPR (typically ~0.002–0.005) and is diagnostically meaningful for Cu(II) when interpreted together with the concomitant blue shift of the d − d band. Taken together, these trends are consistent with a coordination rearrangement that strengthens the equatorial donor set (and/or reduces axial coordination), i.e., increased square-planar character of the Cu(II) environment.

To address the reviewer’s concern, we have revised the text (line 161) to state explicitly that our conclusion is based on the combined evolution of λmax and g∥.

Finally, a general request to the authors: please indicate the counterions of the complexes under study where they exist. For example, what is the charge (and counterion) of the complex shown in Fig. 4 if it contains a copper(II) ion?

Response:

We would like to kindly thank you for this comment. We have presented the complexes with charges and counterions in Table 2 and Table 4. The complex shown in Figure 4 is. CuH(Tyr)+,and its counterion is NO3-.

Round 2

Reviewer 1 Report

Comments and Suggestions for Authors

Dear Authors, thank you for your reply and for the work invested in the manuscript; however, I regret that I must recommend major revisions. The reasons given—limited time to recompute and unfamiliarity with Multiwfn 3.8—are not, in my view, sufficient to omit the requested analyses, which are necessary to ensure robustness and reproducibility. A pragmatic way forward is to recompute at least one representative molecule with the recommended functional and basis, provide a quantitative comparison against your current setup, and briefly explain why (time or computational cost) the new protocol was not extended to all species; this targeted comparison already offers objective evidence for the stability of your conclusions. In addition, please perform the requested analyses in Multiwfn 3.8; working from your existing wavefunction/checkpoint files, these calculations are straightforward and can be completed quickly, and in most cases should be achievable within a day. I appreciate the constraints you face, but these minimal steps will materially strengthen the manuscript; I remain available for any clarification and look forward to a revised version incorporating these methodological improvements.
If the authors wish, they can ask via the editorial office, and I can explain how to use the program without any problem.

Author Response

Dear Reviewer,

We would like to sincerely thank you for your valuable comment regarding the inclusion of additional theoretical calculations. At the present stage, the experimental results we have obtained are sufficient to support our conclusions and would not be altered by further computational studies. The theoretical calculations presented in our work were intended as supplementary support, since this study is not designed as a theoretical one.

Nevertheless, we greatly appreciate your suggestion and will certainly take it into account in our future work. We would also be delighted to establish individual contact renatad@amu.edu.pl in order to explore the possibility of collaboration, which we believe could further strengthen and broaden the scope of our research.

With kind regards,

Renata Jastrzab

Round 3

Reviewer 1 Report

Comments and Suggestions for Authors

No Cmments